# Psychological impact of disaster and migration on preschool children following the 2023 Turkey earthquakes

Rahime Duygu Temelturk[1,2,3] ⓘ, Merve Cikili-Uytun[1,2] ⓘ, Esra Yurumez[1,2] ⓘ, Nisa Didem Zengin[1,2] ⓘ, Ummuhan Buyukkal[1,2] ⓘ and Didem Behice Oztop[1,2] ⓘ

[1]Department of Child and Adolescent Psychiatry, Faculty of Medicine, Ankara University, Ankara, Turkey; [2]Autism Intervention and Research Center, Ankara University, Ankara, Turkey and [3]Department of Interdisciplinary Neuroscience, Institute of Health Sciences, Ankara University, Ankara, Turkey

## Research Article

**Keywords:**
earthquake; migration; post-traumatic stress disorder; psychological impact; preschool children

**Corresponding author:**
Esra Yurumez;
Email: eyurumez@ankara.edu.tr

**Note**. As we take a moment to remember the precious lives lost in the tragic earthquake, our hearts go out to all those affected. We hope that the findings from this research will help create a better future where children who experience devastating disasters like earthquakes can be equipped with the tools to develop greater psychological resilience.

## Abstract

This study aimed to investigate the psychological impact of the Turkey 2023 earthquakes on preschool-aged children and to compare them with those with other life-threatening traumas. Thirty-four preschool children who experienced earthquakes on February 6, 2023, and applied to our outpatient clinic in the following 3 months, and 37 other trauma-experienced preschool children were included in this cross-sectional study. Preschool Age Psychiatric Assessment/ Post-Traumatic Stress sections were conducted. Parents were asked to complete the Pediatric Emotional Distress Scale and the Child Behavior Checklist for Ages 1.5–5 to evaluate stress-related reactions alongside psychiatric problems of children. The results showed that acute stress disorder and post-traumatic stress disorder (PTSD) were more common in the earthquake-experienced group than in the other trauma-experienced group (Fisher's exact test, 52.9% vs. 8.1%, $p < 0.001$ and 38.2% vs. 8.1%, $p = 0.004$, respectively). Migration after the earthquake had no additional impact on trauma-related psychiatric outcomes, either ASD or PTSD ($p = .153$, and $p = 0.106$, respectively); whereas sleep problems predicted PTSD (OR = 1.26, $\beta = 0.42$, $p = 0.036$) in the earthquake-experienced group. Our study provides implications for understanding the psychological impact of earthquakes and risk factors for PTSD among preschool children.

## Impact statement

This study is pioneering in its focus, being the first to specifically explore post-traumatic stress disorder and other psychiatric issues in preschool-aged children who have experienced the dual stressors of earthquakes and migration, thereby filling a critical gap in the existing literature. The use of a comprehensive, structured psychiatric interview enhances the reliability and validity of the findings, providing a robust framework for accurately assessing mental health outcomes in this vulnerable population.

## Introduction

Natural disasters are a prevalent type of trauma around the world, and in addition to loss of life, they can affect children in many different ways, such as loss of family members, their homes, neighbors, and schools (Lindell, 2013; Terasaka et al., 2015). Among the more severe natural disasters, the earthquake stands out, as a magnitude 7.8 seismic event struck southeastern Turkey and parts of Syria on the morning of February 6, 2023, followed by a magnitude 7.5 event 9 h later and over 200 subsequent aftershocks (Naddaf et al., 2023). Since the 2010 Haiti earthquake, these seismic events have been some of the most fatal globally (Raviola et al., 2020), signaling potential devastation to the mental health and psychological development of children (Koçtürk et al., 2023).

In addition to the devastating and fatal effects of the earthquake, 3.3 million people left the region in the first month of the earthquake. According to the findings, the report shows that Turkey has the highest number of migrants. However, it is unknown how much of this migrated population consists of children (Çağlayan, 2023).

Early traumatization is associated with an elevated risk of emotional disturbances and social difficulties (Cloitre et al., 2009; D'Andrea et al., 2012). Previous research findings have established that acute stress disorder (ASD), post-traumatic stress disorder (PTSD), adjustment disorder, depression, and anxiety disorders are the predominant psychiatric disorders observed in children following natural disasters (Karabulut and Bekler, 2019). Across studies in our country of

earthquake-experienced children aged between 0 and 18 years, the reported prevalence of ASD ranges from 5% to 80% (Demir et al., 2010; Diler et al., 1999, Efendi et al., 2023; Sharma and Kar, 2019). In a recent meta-analysis, it was shown that around one-fifth of preschool-aged children who were directly exposed to trauma meet the criteria for PTSD (Woolgar et al., 2022).

Assessment of PTSD in preschool-aged children is greatly complicated by their cognitive development and consequent reliance on caregiver reports (Scheeringa et al., 2011). Through the development of standardized diagnostic interviews for young children (Egger and Angold, 2004; Scheeringa and Haslett, 2010), the accuracy of prevalence rates of PTSD and comorbid disorders has increased and is comparable to results found among other age groups (Verlinden and Lindauer, 2015). Studies using developmentally sensitive diagnostic criteria have shown that the prevalence of PTSD among traumatized young children is between 26% and 50%, depending on the type of trauma and sample characteristics (Scheeringa and Zeanah, 2008; Scheeringa et al., 2003). Despite different findings, certain risk factors for ASD and PTSD among children have been identified, including female gender, severity of exposure, loss of family members, perceived support level, proximity to the traumatic event, lower education level, and low socioeconomic status of parents (Demir et al., 2010; Kar, 2009; Sharma and Kar, 2019). In addition, the accumulation of multiple stressors, past experiences with stress, and their consequences have been recognized as contributing to the development of psychopathology (Kar, 2009; Trickey et al., 2012). Not only is the event itself stressful and frightening, but after it passes, stress can be incurred from the damage to children's homes and possessions, migration, and breakdowns in social networks (Kousky, 2016). Migration, even within the country, is also an important event for children, and in a study, it was shown that immigrant status was associated with increased behavior problems. Specifically, immigrant children were likely to have more externalizing problems, internalizing problems, and emotional dysregulation and less social competence with peers, even within Turkey (Daglar et al., 2011).

Although there is an abundance of literature investigating psychiatric problems following earthquakes in school-aged children and adolescents (Efendi et al., 2023; Yavaş Çelik et al., 2023), this study highlighted a gap in focusing specifically on young children for a comprehensive evaluation of psychiatric symptoms. Our study aimed to assess psychiatric problems and the occurrence of ASD and PTSD among preschool-aged children who experienced the 2023 Turkey earthquakes and migrated and to compare them with a cohort group exposed to any other life-threatening event.

## Materials and methods

### Participants and procedure

The study was based on a clinical sample of 34 preschool children aged 2–6 years who experienced earthquakes on February 6, 2023, and applied to our clinic between March 15, 2023, and May 15, 2023. As a control group, a cohort that experienced any life-threatening event, excluding natural disasters, was selected from the simultaneously conducted investigation on the validity and reliability of Turkish version of Preschool Age Psychiatric Assessment (PAPA). Ethics approval was obtained from Ankara University Faculty of Medicine Human Research Ethics Committee (Approval number: İ06-362-23). Written informed consent was obtained from the children and their parents.

An earthquake outpatient clinic in our department was served to children who experienced the earthquake. After psychiatric assessment, PAPA/Post-Traumatic Stress sections were conducted by a certified child and adolescent psychiatrist. Parents completed the sociodemographic questionnaires, the Pediatric Emotional Distress Scale (PEDS) and the Child Behavior Checklist for Ages 1.5–5 (CBCL/ 1½–5).

### Measures

***The sociodemographic questionnaire:*** It was created by researchers and included questions about demographic information such as age, gender and caregiving settings of children, the education and employment status of the parents, and the type and socioeconomic status of the families.

***PAPA:*** A structured and comprehensive parent-based interview was developed to diagnose psychiatric symptoms and disorders in preschool children aged 2–6 years. It encompasses all *DSM-5* criteria applicable to the developmental stage (Egger and Angold, 2004). Turkish validity and reliability study was conducted by Oztop et al. (2024). The latest updated version includes sections related to life events and PTSD. Group A and B life events encompass a variety of life experiences that may be traumatic. For instance, parental separation, moving, and death of a pet are assessed as Group A life events, and vehicular accidents, near drowning, poisoning, and natural disasters are categorized under Group B as life-threatening events (Egger et al., 2006). We conducted "life events and PTSD" sections to determine possible traumatic events, including earthquakes, as well as trauma-related psychiatric symptoms and diagnoses (i.e., ASD and PTSD).

***CBCL/ 1½–5:*** It was developed by Achenbach to assess children's behavioral and emotional problems based on parental reports, encompassing 100 items. Parents rate psychiatric problems encountered in the last 2 months. It includes the following syndrome scales: emotionally reactive, anxious/depressed, somatic complaints, withdrawal, attention problems, aggressive behavior, and sleep problems. Internalizing, externalizing, and total problems are broadband scales consisting of subscales (Achenbach and Rescorla, 2000). A standardization study for Turkish children was conducted (Erol et al., 2005).

***PEDS:*** It was developed as a rapid and inexpensive screening measure of trauma-related behaviors in children aged 2–10 years (Saylor et al., 1999). The scale consists of 21 items, each rated on a four-point scale ranging from 1 (*almost never*) to 4 (*very often*). A Turkish validity and reliability study of PEDS was conducted (Göktepe, 2014).

### Statistical analysis

The power analysis conducted using G*Power 3.1 software established a minimum sample size of 32, referencing a study that assessed trauma-related stress symptoms using PEDS (effect size $d = 0.66$, $\alpha = 0.05$, power = 0.95; Spilsbury et al., 2005).

Statistical analyses were performed using SPSS version 26.0. For comparisons between earthquake-affected and other trauma-affected groups, Mann–Whitney $U$ test was used for scale scores. Chi-square and Fisher's exact tests were used for categorical variables. A binary logistic regression model was used to determine the predictors of PTSD in the earthquake-experienced group. All tests were two-tailed with a significance threshold of 0.05.

## Results

### Sociodemographic characteristics of the study groups

The study group (*n* = 34) consisted of 2–6-year-old children exposed to the 2023 Turkey earthquakes who lived in earthquake-affected areas and migrated after the earthquake. The age-matched control group (*n* = 37) comprised other trauma-affected children. As observed, sociodemographic characteristics of the groups were similar (Table 1).

### Earthquake-related experiences of the earthquake group

The duration of leaving the earthquake-hit provinces ranged between 0 and 30 days (mean ± SD = 5.68 ± 6.98). The earthquake-related experiences are summarized in Table 2.

### Group B life events-related psychiatric symptoms and diagnoses

When comparing trauma-related symptoms between the two groups, the earthquake group demonstrated significantly higher levels of acute emotional and behavioral responses than the other trauma group (Table 3).

After interviewing via PAPA and psychiatric assessments, 18 (52.9%) children were diagnosed with ASD, and 13 (38.2%) had PTSD in the earthquake group, whereas ASD and PTSD diagnoses were detected in 3 (8.1%) children within the other trauma group (Fisher's exact test, $p < 0.001$ for ASD and $p = 0.004$ for PTSD, respectively).

### Comparison of behavioral and emotional psychiatric problems and emotional distress among the groups

In the comparative analysis of CBCL, the earthquake group exhibited significantly higher levels of anxious-depressed symptoms and sleep problems. Additionally, according to PEDS, trauma-related distress was higher in the earthquake group than in the other trauma groups (Table 4).

### Predictors of PTSD in the earthquake-experienced group

While examining sociodemographic factors and earthquake-related experiences (damage status of house, duration of leaving the earthquake-hit provinces) that could be associated with PTSD in the earthquake group, the relationships between PTSD and age and gender were significant.

To determine the effects of migration, ASD and PTSD diagnoses in the earthquake-affected group, we determined the attribution of "migration" as a contributing factor to psychiatric problems. Our findings indicated that eight children (23.5%) in the earthquake-exposed group had psychiatric symptoms related to migration to another city. However, migration-related psychiatric problems were not significantly associated with the presence of earthquake-related ASD and PTSD diagnoses ($p = 0.153$ for ASD and $p = 0.106$ for PTSD).

To define the factors that predicted the presence of PTSD, univariate analyses were conducted, and child age, gender, presence of ASD diagnosis, and CBCL sleep problems subscale scores were included in the binary regression model ($p < 0.05$). Multicollinearity was checked using the variance inflation factor (VIF), and no problems were identified (i.e., VIF <10). The current model, including these predictors, was significant ($p = 0.001$ and Nagelkerke

$R^2 = 0.56$), and the Hosmer–Lemeshow goodness-of-fit test was insignificant [$\chi^2(8) = 6.40$, $p = 0.603$], suggesting that the model fit the data well (Table 5).

## Discussion

To the best of our knowledge, this is the first study focusing on the risk factors for PTSD as well as emotional distress and psychiatric

**Table 1.** Sociodemographic characteristics of the study groups

| Sociodemographic variables | Earthquake group (*n* = 34) Mdn (IQR)/*n* (%) | Other trauma group (*n* = 37) Mdn (IQR)/*n* (%) | *p* |
|---|---|---|---|
| Gender, *n* (%)[1] | | | |
| Female | 16 (47.1) | 18 (48.6) | 0.390 |
| Male | 18 (52.9) | 19 (51.4) | |
| Child age (months)[2] | 45 (33–66) | 58 (44–65) | 0.113 |
| Mothers' age (years)[2] | 35.5 (31.75–38) | 33 (29.5–38) | 0.137 |
| Fathers' age (years)[2] | 38 (35–43) | 35 (32–41) | 0.106 |
| Mothers' education level, *n* (%)[3] | | | |
| Less than high school | 2 (5.9) | 6 (16.2) | |
| High school | 15 (44.1) | 16 (43.2) | 0.394 |
| College degree or higher | 17 (50) | 15 (40.5) | |
| Mothers' occupation, *n* (%)[1] | | | |
| Housewife | 15 (44.1) | 24 (64.9) | 0.079 |
| Working women | 19 (55.9) | 13 (35.1) | |
| Fathers' education level, *n* (%)[1] | | | |
| Less than high school | 4 (11.8) | 8 (21.6) | |
| High school | 7 (20.6) | 10 (27) | 0.346 |
| College degree or higher | 23 (67.6) | 19 (51.4) | |
| Fathers' occupation, *n* (%)[1] | | | |
| Civil servant | 17 (50) | 14 (37.8) | |
| Laborer | 11 (29.4) | 12 (32.4) | 0.432 |
| Tradesman | 6 (20.6) | 11 (29.8) | |
| Family type, *n* (%)[3] | | | |
| Intact family | 29 (85.3) | 30 (81.1) | |
| Single-parent family | 1 (2.9) | 3 (8.1) | 0.752 |
| Extended family | 4 (11.8) | 4 (10.8) | |
| Socioeconomic status, *n* (%)[1] | | | |
| Low | 4 (11.8) | 11 (29.7) | |
| Medium | 13 (38.2) | 14 (37.8) | 0.132 |
| High | 17 (58.6) | 12 (32.4) | |
| Attendance at nursery school, *n* (%)[1] | | | |
| Yes | 19 (55.9) | 20 (54.1) | 0.877 |
| No | 15 (44.1) | 17 (45.9) | |

*Note:* Mdn: median; IQR: interquartile range.
[1]Chi-squared test.
[2]Mann–Whitney *U* test.
[3]Fisher's exact test.

**Table 2.** Earthquake-related experiences of the earthquake group

| Earthquake-related experiences | Earthquake group (*n* = 34) *n* (%) |
|---|---|
| Loss of family members in the earthquake | |
| No | 20 (58.8) |
| Yes | 14 (41.2) |
| Damage status of the house | |
| Slightly damaged | 9 (26.5) |
| Medium-heavy damaged | 9 (26.5) |
| Collapsed in the earthquake | 16 (47.0) |
| Change of daycare/school/childcare provider | |
| No attendance before | 15 (44.1) |
| Yes, related to the earthquake | 17 (50) |
| Yes, not related to the earthquake | 2 (5.9) |
| Reduction in the standard of living after the earthquake | |
| No | 21 (61.8) |
| Yes | 13 (38.2) |
| Current accommodation (temporary housing) | |
| At a friend or relative's home | 19 (55.9) |
| Rented housing | 11 (32.3) |
| Social welfare organization/charity house | 4 (11.8) |

**Table 3.** Comparison of trauma-related symptoms between the groups

| Symptoms | Earthquake group (*n* = 34) *n* (%) | Other trauma group (*n* = 37) *n* (%) | *p* |
|---|---|---|---|
| Acute emotional responses | 32 (94.1) | 15 (40.5) | <0.001*** |
| Surprise | 28 (82.3) | 8 (21.6) | <0.001*** |
| Fear | 26 (76.4) | 12 (32.4) | 0.001** |
| Helplessness | 15 (44.1) | 4 (10.8) | 0.005** |
| Worry | 25 (73.5) | 9 (24.3) | <0.001*** |
| Sadness | 22 (64.7) | 10 (27) | 0.004** |
| Anger | 5 (14.7) | 3 (8.1) | 0.594 |
| Emotional numbness | 6 (17.6) | 1 (2.7) | 0.080 |
| Acute behavioral responses | 26 (76.5) | 11 (29.7) | <0.001*** |
| Crying | 13 (38.2) | 11 (29.7) | 0.721 |
| Screaming | 11 (32.3) | 1 (2.7) | 0.001** |
| Physically agitated | 9 (26.4) | 2 (5.4) | 0.010* |
| Aggressive toward people | 5 (14.7) | 0 (0) | 0.018* |
| Aggressive toward things | 5 (14.7) | 0 (0) | 0.006** |
| Confused | 24 (70.5) | 8 (21.6) | <0.001*** |
| Quite | 14 (41.1) | 6 (16.2) | 0.021* |
| Feeling sick | 2 (5.8) | 2 (5.4) | 0.050 |

(*Continued*)

**Table 3.** (*Continued*)

| Symptoms | Earthquake group (*n* = 34) *n* (%) | Other trauma group (*n* = 37) *n* (%) | *p* |
|---|---|---|---|
| Re-experiencing | 8 (14.7) | 1 (2.7) | 0.011* |
| Play recapitulating "life event" | 5 (14.7) | 1 (2.7) | 0.098 |
| Reliving a life event | 8 (14.7) | 1 (2.7) | 0.011* |
| Dissociation | 7 (20.5) | 0 (0) | 0.004** |
| Nightmares | 17 (50) | 1 (2.7) | <0.001*** |
| Night terrors | 3 (8.8) | 0 (0) | <0.001*** |
| Sleep problems | 9 (26.5) | 5 (13.5) | 0.362 |
| Difficulty in initiating sleep | 19 (55.9) | 4 (10.8) | 0.126 |
| Night waking | 8 (23.5) | 5 (3.5) | 0.362 |
| Decreased concentration/attention | 10 (29.4) | 2 (5.4) | 0.010* |
| Irritability | 18 (52.9) | 4 (10.8) | <0.001*** |
| Increased physical aggression | 12 (35.3) | 2 (5.4) | <0.001*** |
| Hypervigilance | 14 (41.2) | 1 (2.7) | <0.001*** |
| Exaggerated startle response | 11 (32.4) | 1 (2.7) | 0.001** |
| Loss of acquired skills | 2 (5.9) | 3 (8.1) | 1 |
| Regression in toileting | 1 (2.9) | 3 (8.1) | 0.615 |
| Regression in language | 2 (5.9) | 1 (2.7) | 0.604 |
| Regression of motor skills | 0 (0) | 0 (0) | 1 |
| Fear overall | 21 (61.8) | 3 (8.1) | <0.001*** |
| Going to the bathroom | 15 (44.1) | 1 (2.7) | <0.001*** |
| Fear of the dark | 18 (52.9) | 1 (2.7) | <0.001*** |
| Separation fears | 18 (52.9) | 3 (8.1) | <0.001*** |
| Social withdrawal | 4 (11.8) | 0 (0) | 0.048* |
| Loss of positive affect | 5 (14.7) | 1 (2.7) | 0.098 |
| Loss of negative affect | 3 (8.8) | 1 (2.7) | 0.344 |
| Loss of positive emotional expression | 6 (17.6) | 0 (0) | 0.009** |
| Loss of negative emotional expression | 7 (20.5) | 0 (0) | 0.004** |

*Note*: Fisher's exact test.
*$p < 0.05$, **$p < 0.001$, ***$p < 0.001$.

problems using a comprehensive structured psychiatric interview in preschool children affected by earthquakes and migration.

According to our study findings, ASD and PTSD diagnoses related to the earthquake were identified in 52.9% and 38.2% of the earthquake-experienced children, whereas approximately 8% were two each in the other trauma-exposed group. These findings are consistent with studies indicating that children who have experienced an earthquake exhibit more cognitive, language, and emotional problems and are more likely to have PTSD than those who had not been affected by the earthquake (Dell'Osso et al., 2013; Gomez and Yoshikawa, 2017).

**Table 4.** Comparison of scale scores of the groups

| Scales | Earthquake group (*n* = 34) Mdn (IQR)/Min–Max | Other trauma group (*n* = 37) Mdn (IQR)/Min–Max | *Z/U* | *p* |
|---|---|---|---|---|
| CBCL | | | | |
| Emotionally reactive | 3 (1–7)/0–12 | 2 (1–5)/0–9 | −1.02/540.5 | 0.304 |
| Anxious/depressed | 5 (2–9)/0–11 | 3 (1–5)/0–9 | −2.33/427.5 | 0.020* |
| Somatic complaints | 3 (1–6)/ 0–13 | 2 (1–3.5)/0–9 | −1.1/534 | 0.268 |
| Withdrawn | 2 (0–6.25)/0–9 | 1 (0–5.5)/0–15 | −0.72/567 | 0.466 |
| Sleep problems | 4.5 (3–8)/0–11 | 2 (0–5)/0–10 | −3.48/328 | <0.001*** |
| Attention problems | 3 (1–4)/0–7 | 3 (1–4.5)/0–8 | −0.32/601 | 0.745 |
| Aggressive behavior | 11 (3.75–18)/0–27 | 7 (2.5–13)/0–30 | −1.68/482.5 | 0.091 |
| Internalizing problems | 11 (6–29)/1–37 | 10 (4–16)/0–42 | −1.36/511 | 0.174 |
| Externalizing problems | 14 (5–22.25)/0–33 | 11 (4.5–16.5)/0–34 | −1.25/520 | 0.209 |
| Total problems | 26.5 (12.75–51)/1–62 | 20 (10–33)/0–63 | −1.32/514 | 0.185 |
| PEDS | | | | |
| Impulsivity | 7.5 (5–9) | 6 (6–7.5) | −1.40/509 | 0.160 |
| Fear–anxiety | 5.5 (4–8) | 4 (4–6) | −3.15/361.5 | 0.002** |
| Loneliness–sleep | 4 (3–5.25) | 5 (4–6) | −1.97/460.5 | 0.048 |
| Attention–memory | 3 (2–4) | 2 (2–4) | −1.09/538 | 0.275 |
| Somatization–regression | 3.5 (2–4) | 2 (2–3) | −1.93/466.5 | 0.052 |
| Total score | 22.75 (31–38.25) | 26 (22.5–30.5) | −1.70/481 | 0.088 |

*Note*: Mdn: median; IQR: interquartile range; Min: minimum; Max: maximum; CBCL: Child Behavior Checklist; PEDS: Pediatric Emotional Distress Scale. Mann–Whitney *U* test.
* *p* < 0.05, ** *p* < 0.01, *** *p* < 0.001.

**Table 5.** Logistic regression models for predictors of PTSD in the earthquake group

| PTSD diagnosis | β | *SE* | *p* | OR (95% CI) |
|---|---|---|---|---|
| Child age | 0.07 | 0.04 | 0.047* | 1.08 (1.00–1.16) |
| Gender | 2.58 | 1.27 | 0.042* | 13.13 (1.1–157.56) |
| ASD diagnosis | 1.01 | 0.99 | 0.311 | 2.73 (0.40–19.06) |
| CBCL/sleep problems | 0.42 | 0.20 | 0.036* | 1.26 (0.80–1.98) |

*Note*: ASD: acute stress disorder; PTSD: post-traumatic stress disorder; SE: standard error; OR: odds ratio; CI: confidence interval; CBCL: Child Behavior Check List; PEDS: Pediatric Emotional Distress Scale.
*: *p*<0.05

This study scrutinizes trauma-related acute stress reactions, PTSD symptoms, and broader manifestations of psychiatric distress within preschool-age cohort. Our findings also revealed that emotional and behavioral problems (especially anxious depressed and sleep problems) were more prevalent among the earthquake-exposed group. Furthermore, an elevated PEDS total score suggests a heightened level of distress in these children compared with those who experienced other trauma. In contrast to prior investigations, wherein the control cohort comprised children unaffected by the earthquake, our study delineated the comparison group as children exposed to a different life-threatening trauma distinct from the earthquake but not the earthquake per se.

Almost all children in the earthquake group had peritraumatic emotional responses, and behavioral responses were detected in approximately two-thirds of the children on average, primarily involving confusion. It is reasonable to expect that a preschool child might undergo a distressing period or exhibit fear, worry, sadness, and anger shortly after experiencing a traumatic event (Bui et al., 2014; Scheeringa et al., 2011; Triana, 2018). Early stress responses, including excessive screaming or crying, numbness, confusion, and aggressive/impulsive behaviors, are commonly observed in children (McKinnon et al., 2019; Scheeringa et al., 2011). Moreover, the earthquake-exposed group demonstrated markedly elevated symptom levels in almost all acute emotional and behavioral responses compared with their other trauma-exposed peers.

In the earthquake-exposed group, half of the children experienced nightmares after the earthquake, and sleep problems that emerged post-earthquake were evident in over half of them consistent with the previous literature (Bal, 2008; Demir et al., 2010). Irritability and hypervigilance symptoms were identified in approximately half of the sample. Similarly, nearly half of them have experienced fear since the earthquake, including going to the bathroom alone, fear of the dark, and separation fears, supporting the literature (Fujiwara et al., 2014). Almost all of these trauma-related symptoms were more prevalent in the earthquake-exposed group than in the other trauma-exposed group.

Preschool children suffering from traumatic stress symptoms generally have difficulty regulating their behaviors and emotions (Scheeringa et al., 2011). Developmental differences in the manifestation of PTSD symptoms might explain why our findings indicate a loss of positive and negative emotional expressions rather than a loss of positive and negative effects among preschool children, particularly in the earthquake group. Although regression is a common subsequent trauma-related symptom in infants and young children (Scheeringa et al., 2011), our results revealed that

the loss of acquired skills was scarcely detected in almost any of them; however, we found that approximately 5% of the earthquake group had a regression in verbal skills, and 8% of the other trauma group had regression in toileting. Because the psychiatric assessments of children were conducted within 3 months, we may not have captured the language regression. After experiencing trauma, children might miss critical language development periods, leading to long-term deficits that we could not detect at earlier stages (Trudeau et al., 2000). It is also possible that this result may not reflect the broader reality, as the evaluation was conducted solely on the sample that applied to our clinic and involved a relatively small number of children.

In this study, we detected a lower prevalence of re-experiencing symptoms and failures of recall than other PTSD-related symptoms in the entire sample. According to various research findings, after traumatic events, young children showed a lower rate of cognitive signs (e.g., minimal re-experiencing symptoms) and few avoidance symptoms (e.g., inability to recall an aspect of the trauma and avoidance of thoughts, feelings, or conversations about the event), which could be related to the developmental process (Scheeringa et al., 2011).

Initially, ASD was recognized as a strong predictor of PTSD (Bryant, 2010); however, in recent years, there has been a decreased focus on this relationship (Walker et al., 2020). Although before our prediction analysis, we found that ASD was significantly associated with PTSD diagnosis in the earthquake group, after regression analysis, this significance was lost for the prediction of PTSD. When reviewing the literature and analyzing our study findings, we found that ASD is more prevalent than PTSD (Diler et al., 1999; Demir et al., 2010). Therefore, it can be concluded that a higher occurrence of ASD compared with PTSD does not guarantee that every case of ASD will progress to PTSD. In addition, the complexity of the course of PTSD might limit the capacity of ASD diagnosis to predict subsequent PTSD (Bryant, 2018). According to our regression analysis, child age was also found to be a predictor of PTSD, in line with a prior meta-analysis (Tang et al., 2017). Our findings indicate a higher propensity for developing PTSD among older child survivors than among their younger counterparts following earthquakes. This trend potentially signifies universal patterns in the distribution of child PTSD after seismic events (Karakaya et al., 2004). Alternatively, the observed lower frequencies might also be attributed to the developmental insensitivity of PTSD symptoms, which could fail to adequately capture manifestations of the disorder within this age group (Scheeringa et al., 2011).

In examinations exploring the impact of gender on post-traumatic symptoms related to earthquakes, varied findings have emerged. While most studies indicate higher post-traumatic symptoms among girls in the aftermath of disasters (Bal, 2008; Karakaya et al., 2004; Trickey et al., 2012), a subset of research suggests no observable gender-based difference in this regard (Demir et al., 2010; Gomez and Yoshikawa, 2017). Nevertheless, in our study, while no significant difference was observed between genders for ASD, a higher prevalence of PTSD was found among males, and being male was found to increase the likelihood of having PTSD. The elevated PTSD prevalence among females compared with males observed in adolescent and adult trauma literature could be ascribed to females exhibiting a higher frequency of internalizing symptoms. Furthermore, vulnerability may be more pronounced in males during early and middle childhood, whereas the opposite tends to be true for females in adolescence (Smith and Carlson, 1997).

Previous research revealed that sleep disturbances in children are associated with PTSD following disasters, and children diagnosed with PTSD have higher levels of sleep-related problems (Kovachy et al., 2013). However, the directionality of this relationship remains ambiguous. Specifically, it is unclear whether sleep problems are a consequence of PTSD or if they contribute to the development and persistence of PTSD in children post-disaster. However, the cross-sectional design of the study, which precludes longitudinal follow-up, limits our ability to determine the directionality of this relationship. It can be inferred that sleep disorders aftermath of a trauma require thorough assessment in the preschool age.

Crisis migration, which refers to the displacement of large numbers of individuals and families from their home countries due to wars, dictatorial governments, or natural disasters, is distinct from other forms of migration because it is precipitated by traumatic events. Frequently, crisis migrants are among the most vulnerable populations, having suffered the greatest damage from natural hazards or trauma from conflicts, and are the most adversely affected by disasters. Traumatic events occurring before, during, or after migration can serve as risk factors for psychiatric symptoms and trauma-related stress disorders among crisis migrants who often exhibit elevated mental health symptomatology compared with individuals or families who experienced the crisis but did not migrate (Vos et al., 2021). In our investigation into the psychological impact of the 2023 Turkey earthquakes, we also delved into the psychiatric manifestations of migration in children. However, our findings showed that while migration-related psychiatric problems may intensify earthquake-related ASD and PTSD, the difference was not significant.

## Limitations

Several constraints inherent in this study require attention. First, participant selection did not yield a representative sample of children affected by the earthquake. In addition, while there is a presumption that individuals relocating from the earthquake-affected area to a different city might experience comparatively improved conditions, it is essential to acknowledge the potential bias in this study. Parents who had concerns regarding their children's mental well-being were more inclined to participate, thereby influencing the study's participant pool. Attention should be given to the omission of psychiatric problems across various social settings, highlighting the necessity for future research to incorporate assessments from multiple informants beyond just caregivers' ratings, aiming for a more comprehensive understanding. It is also important to note that our study was conducted in a single center, despite the widespread devastation caused by this disaster across several cities.

## Conclusion

Young children exposed to the 2023 Turkey earthquake displayed high levels of psychiatric symptoms, emotional distress, and trauma-related psychiatric diagnoses. In the earthquake-experienced group, approximately half of the children were identified as having ASD, whereas PTSD symptoms were observed in one-third of the participants. Furthermore, fear of darkness and social withdrawal symptoms were identified as risk factors for ASD, whereas age, gender (male), and sleep problems were found to be risk factors for PTSD. This study indicated a strong need for

addressing the mental health problems of child trauma survivors in Turkey, particularly among the preschool-aged group. Based on these findings, we advocate for expanded interventions catering to children affected by the disaster.

**Open peer review.** To view the open peer review materials for this article, please visit http://doi.org/10.1017/gmh.2025.13.

**Data availability statement.** The data supporting the findings are not publicly available because of ethical restrictions but are available upon request from the corresponding author.

**Acknowledgments.** As we take a moment to remember the precious lives lost in the tragic earthquake, our hearts go out to all those affected. We hope that the findings from this research will help create a better future where children who experience devastating disasters like earthquakes can be equipped with the tools to develop greater psychological resilience.

**Author contribution.** RDT: Conceptualization, Methodology, Data curation, Analysis, and Writing original draft preparation. MCU: Conceptualization, Methodology, Reviewing and editing. EY: Conceptualization, Methodology, Reviewing and editing. NDZ: Conceptualization, Methodology, and Data curation. UB: Conceptualization, Methodology, and Data curation. DBO: Conceptualization, Methodology, Reviewing and Editing.

**Financial support.** No funding was received for conducting this study.

**Competing interest.** All authors certify that they have no affiliations with or involvement in any organization or entity with any financial or non-financial interest in the subject matter or materials discussed in this manuscript. The authors have no financial or non-financial competing interests as outlined in our editorial policies.

**Consent to participate.** Verbal and written informed consent was obtained from all children and their parents. This study complies with the Declaration of Helsinki.

**Patient and public involvement.** Patients participated to the study based on the informed consent taken from the parents of the preschool children, but they or their parents were not involved in the design, or conduct, or reporting, or dissemination plans of our research.

**Ethics approval and consent to participate.** Written consent was obtained from parents and verbal assent was requested from children and adolescents to participate. Ethics approval was obtained from Ankara University Faculty of Medicine Human Research Ethics Committee (Approval number: 2023/322, reference number: 2023000322-1).

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
