## [Reviewer Report]

The manuscript addresses a highly significant and timely topic by exploring the psychological impact of the 2023 Turkey earthquakes and migration on preschool children. The study has the potential to make meaningful contributions to the literature on post-disaster interventions and migration processes. However, certain methodological details and the discussion section could benefit from further clarity and depth to enhance the overall impact of the paper.

Some sentences and expressions appear to be ambiguous or overly complex, which may hinder the reader’s ability to fully understand the content. The manuscript would greatly benefit from a comprehensive review of its English language to ensure clarity, grammatical accuracy, and overall coherence.

Secondly, the introduction section of the manuscript appears to lack coherence and organization in terms of logical flow and overall structure. This impacts the clarity of the narrative and makes it challenging for readers to follow the context and objectives of the study. A more structured and cohesive approach is recommended to ensure a clear connection between the background information, the research gap, and the study’s aims.

The characteristics of the control group in the Methods section need to be described in greater detail. Providing more comprehensive information about the demographic and clinical features of the control group would enhance the clarity and robustness of the study’s methodology. Additionally, the Methods section states that written consent was obtained from younger children, which raises questions about how this was achieved. Since obtaining written consent from small children is generally not feasible due to their developmental stage, further clarification is needed. Furthermore, you mentioned that a special clinic within your institution provided services to children affected by the earthquake. Additional information about this implementation would be valuable. For instance, it would be helpful to clarify who conducted the psychological assessments of these children and when this clinic was established. Additionally, the manuscript states that applications between 5 March 2023 and 15 May 2023 were included in the study. Was there a rationale for selecting this specific time frame?

In addition, regarding the measures section, it is recommended to provide more detailed information about the Pediatric Emotional Distress Scale (PEDS). For instance, what is the cutoff score used to identify clinically significant distress? Additionally, please specify the time frame the scale assesses, such as symptoms occurring over the past few months. Including these details will enhance the clarity and utility of the methodology.

In the Results section, it is suggested to provide a more detailed description of the sociodemographic and trauma-related characteristics of the control group. Is the time elapsed since the mentioned traumas experienced by the control group known?

Regarding the Discussion part, the practical implications of the findings could be further explored, particularly in relation to psychosocial support programs for children. Moreover, the lack of detailed information about the trauma history of the control group should be acknowledged as a limitation. Clarifying how much time has elapsed since the control group’s trauma exposure would enhance the study’s validity and allow for a more accurate interpretation of the findings. The fact that the parents of children who experienced the earthquake were also exposed to the same trauma also raises an important perspective. The psychological impact on parents might have influenced the differences observed between groups, as parental mental health plays a critical role in shaping children’s emotional and behavioral responses to trauma. Including this perspective in the discussion could enrich the study by addressing how the parents’ psychological well-being may have moderated or amplified the effects observed in their children.

Additionally, it would be valuable to clarify whether trauma experiences related to migration were assessed in the group that migrated after the earthquake. Exploring individual migration-related traumatic events, such as loss of social support, exposure to unsafe conditions, or difficulties during the displacement process, could provide a deeper understanding of the unique psychological challenges faced by this group. Including such an analysis would strengthen the study by addressing potential interactions between disaster- and migration-related traumas.

Future research directions should also be included in the discussion to provide a broader perspective on the study’s findings.

---

## [Reviewer Report]

This is an excellent research article that demonstrates a thorough and comprehensive exploration of the topic. The depth of analysis and the breadth of coverage are commendable, showcasing the author’s strong grasp of the subject matter. The discussion section is particularly well-articulated, providing a detailed interpretation of the findings while effectively situating them within the broader context of existing literature.

The methodology is robust and clearly outlined, and ensuring the study’s credibility.

Overall, this article meets the high standards required for publication in this journal. It is an impressive piece of scholarly work, and I have no hesitation accepting it for publication.

---

## [Reviewer Report]

The authors have adequately addressed the reviewers' comments and made the necessary revisions to improve the manuscript. The responses are comprehensive, and the revised version meets the journal’s standards for clarity, methodology, and scientific rigor. Given these improvements, I believe the manuscript is now suitable for publication.